Extended Abstract Track

# The Union of Manifolds Hypothesis

**Bradley C.A. Brown**[*]                    BCABROWN@UWATERLOO.CA
*University of Waterloo*

**Anthony L. Caterini**                         ANTHONY@LAYER6.AI
**Brendan Leigh Ross**                       BRENDAN@LAYER6.AI
**Jesse C. Cresswell**                           JESSE@LAYER6.AI
**Gabriel Loaiza-Ganem**                     GABRIEL@LAYER6.AI
*Layer 6 AI*

**Editors:** Sophia Sanborn, Christian Shewmake, Simone Azeglio, Arianna Di Bernardo, Nina Miolane

## Abstract

The *manifold hypothesis* states that low-dimensional manifold structure exists in high-dimensional data, which is strongly supported by the success of deep learning in processing such data. However, we argue here that the manifold hypothesis is incomplete, as it does not allow any variation in the intrinsic dimensionality of different sub-regions of the data space. We thus posit the *union of manifold hypothesis*, which states that high-dimensional data of interest comes from a union of disjoint manifolds; this allows intrinsic dimensionality to vary. We empirically verify this hypothesis on image datasets using a standard estimator of intrinsic dimensionality, and also demonstrate an improvement in classification performance derived from this hypothesis. We hope our work will encourage the community to further explore the benefits of considering the union of manifolds structure in data.

## 1. Introduction

The manifold hypothesis (Bengio et al., 2013) states that high-dimensional data of interest often lives in an unknown lower-dimensional manifold embedded in ambient space. Recently, there has been increased interest both in confirming this hypothesis and understating its role in deep learning. Pope et al. (2021) conduct a comprehensive study estimating the intrinsic dimension of commonly-used image datasets, unambiguously finding low-dimensional structure, and linking this structure to the performance of classifiers. Ansuini et al. (2019) conduct a similar study linking the estimated intrinsic dimension of deep-learning-based representations to classification accuracy. The implications of the manifold hypothesis in deep learning are not limited to the supervised setting, extending to unsupervised tasks such as clustering (Elhamifar and Vidal, 2011, 2012, 2013; Abdolali and Gillis, 2021) and generative modelling (Dai and Wipf, 2019; Brehmer and Cranmer, 2020; Arbel et al., 2021; Kothari et al., 2021; Caterini et al., 2021; Ross and Cresswell, 2021; Loaiza-Ganem et al., 2022; Ross et al., 2022). Understanding the symmetries and invariances present in data and deep learning models is a fundamental problem (Bronstein et al., 2021), and studying the nature of the low-dimensional structure present in data is directly related.

However, thinking of observed data as lying on a single unknown low-dimensional manifold is quite limiting, as this immediately implies that the intrinsic dimension throughout

---

[*] Work done during an internship at Layer 6 AI.

the dataset is constant. If we consider the intrinsic dimensionality to be the number of factors of variation generating the data, we can see that this formulation prevents distinct regions of the data's support from having differing factors of variation. Yet this seems to be unrealistic: for example, we should not expect the number of factors needed to describe 8s and 1s in the MNIST dataset (LeCun et al., 1998) to be equal. In other words, while a good first approximation, we should not expect the manifold hypothesis to fully explain the low-dimensional structure commonly found in data. To accommodate this intuition, we propose the *union of manifolds* hypothesis.

We posit that high-dimensional data often lies not on a single manifold, but on a disjoint union of manifolds of *different intrinsic dimensions*. Note that the disjoint union of $d$-dimensional manifolds is itself a $d$-dimensional manifold: the possibility of having different intrinsic dimensions differentiates the union of manifolds hypothesis from the manifold hypothesis. We empirically verify the union of manifolds hypothesis and relate it to classification accuracy.

## 2. Background and Related Work

In order to verify the union of manifolds hypothesis, we rely on estimators of intrinsic dimension. These estimators assume that the support of the data generating distribution is a $d$-dimensional manifold – where $d < D$, and $D$ is the dimension of the ambient space, $\mathbb{R}^D$ – and return an estimate of $d$ based on observed data $\{x_i\}_{i=1}^n$. In their empirical study of intrinsic dimension of commonly-used datasets, Pope et al. (2021) use the Levina and Bickel (2004) estimator with the MacKay and Ghahramani (2005) extension, given by

$$\hat{d}_k := \left( \frac{1}{n(k-1)} \sum_{i=1}^n \sum_{j=1}^{k-1} \log \frac{T_k(x_i)}{T_j(x_i)} \right)^{-1}, \tag{1}$$

where $T_j(x)$ is the Euclidean distance from $x$ to its $j^{\text{th}}$-nearest neighbour in $\{x_i\}_{i=1}^n \setminus \{x\}$, and $k$ is a hyperparameter specifying the number of nearest neighbours to consider (this hyperparameter can be intuitively understood as the scale at which the manifold is probed). While other estimators have been recently proposed (Block et al., 2021; Lim et al., 2021; Tempczyk et al., 2022), we stick with (1) throughout this work as it is well-established in the literature.

We also point out that while the union of manifolds hypothesis has been mentioned before (Elhamifar and Vidal, 2011, 2012, 2013; Abdolali and Gillis, 2021; Cunningham et al., 2022), to the best of our knowledge an empirical verification of it analogous to the one of Pope et al. (2021) (with the manifold hypothesis) has not been carried out before.

## 3. The Union of Manifolds Hypothesis

### 3.1. The Hypothesis

In this section we state the *union of manifolds hypothesis*, which is the main conceptual contribution of this work:

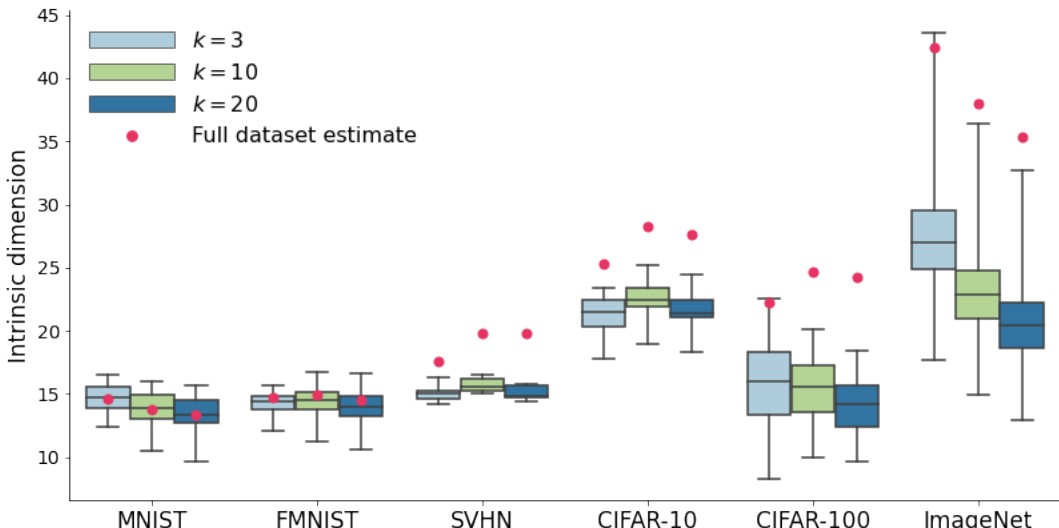

Figure 1: Boxplots showing variability of intrinsic dimension estimates across classes. Each boxplot shows the distribution (minimum, lower quartile, median, upper quartile, and maximum) over the estimated intrinsic dimension of classes in the given dataset (x-axis) for the specified $k$-value (color).

> **Hypothesis:** The support of the data-generating distribution can be written as $\bigsqcup_{\ell=1}^{L} \mathcal{M}_\ell \subset \mathbb{R}^D$, where $\sqcup$ denotes disjoint union, and each $\mathcal{M}_\ell$ is a connected manifold of dimension $d^{(\ell)}$.

Assuming all $d^{(\ell)}$ are identical recovers the standard manifold hypothesis.

### 3.2. Verifying the Hypothesis

In this section we show that commonly-used image datasets are better described by the *union* of manifolds hypothesis than just the manifold hypothesis. In order to achieve this, we first attempt to identify the connected components $\mathcal{M}_\ell$ of the data, and then obtain estimates $\hat{d}_k^{(\ell)}$ of the intrinsic dimension of each connected component through (1) for each $\ell = 1, \ldots, L$. Observing varying estimates across different $\ell$s would strongly support the union of manifolds hypothesis, whereas observing similar estimates would support the manifold hypothesis.

Most image datasets are accompanied by class labels. We argue that class labels are a natural proxy for identifying the connected components $\mathcal{M}_\ell$, as one can easily conceive of continuously deforming any natural image from a given class into another image from the same class without any intermediate image not belonging to the class. For example, on MNIST we can likely continuously transform any 9 into any other 9 without leaving the manifold of 9s, whereas attempting to similarly transform a 9 into a 3 would likely result in intermediate images that are neither 9s nor 3s.

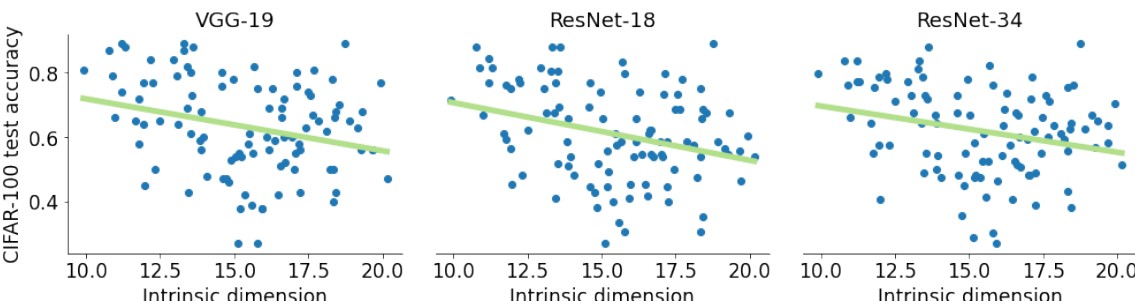

Figure 2: Class intrinsic dimension versus test accuracy on CIFAR-100, along with a least-squares regression line. Correlation coefficients for each model are $-0.243 \pm 0.015$, $-0.269 \pm 0.012$, and $-0.274 \pm 0.007$, respectively (means and standard errors over 5 runs). Corresponding $p$-values, $0.008 \pm 0.002$, $0.019 \pm 0.008$, and $0.006 \pm 0.001$, show the relationship is significant.

Figure 1 shows, for different values of the hyperparameter $k$, boxplots of the values $\hat{d}_k^{(\ell)}$ for class labels $\ell = 1, \ldots, L$ in our considered datasets: MNIST, FMNIST (Xiao et al., 2017), SVHN (Netzer et al., 2011), CIFAR-10, CIFAR-100 (Krizhevsky et al., 2009), and ImageNet (Russakovsky et al., 2015). Two relevant patterns emerge. First, within each dataset, results are consistent across different choices of $k$, so that any conclusions we draw are not caused by a specific choice of this hyperparameter. ImageNet is a partial exception to this consistency across $k$, likely since it has fewer datapoints per class, which can result in (1) underestimating the true intrinsic dimension (Pope et al., 2021). While CIFAR-100 also has fewer datapoints per class than other datasets, consistency across $k$ suggests this is less of a concern in this case. Second, for all datasets except SVHN, we can observe a wide range of estimated intrinsic dimensions. We show more granular estimates in Appendix A, with further choices of $k$ and the intrinsic dimensions of individual classes. These results empirically verify that the union of manifolds hypothesis is more appropriate for images than the manifold hypothesis. For completeness, we also include estimates when using the full dataset in Figure 1, although we point out that these estimates should not be expected to match the average of class-specific estimates, as is clear by inspecting (1).

### 3.3. Intrinsic Dimension and Classification Accuracy

We now study how the union of manifold hypothesis relates to the performance of classifiers. We train 3 classifiers on CIFAR-100: a VGG-19 (Simonyan and Zisserman, 2015), a ResNet-18, and a ResNet-34 (He et al., 2016). We focused on CIFAR-100 here as the other datasets considered in this work were either too simple to classify (MNIST, FMNIST, SVHN, CIFAR-10), or produced less-reliable intrinsic dimension estimates (ImageNet). For each class, we compute its classification accuracy and plot this against its estimated intrinsic dimension in Figure 2. We can see that, consistently across classifiers, there is an inverse relationship between estimated intrinsic dimension and classification accuracy. We also compute the correlation between these two quantities for a quantitative comparison, and compute a $p$-

Table 1: Means and standard errors of ResNet-18 accuracy on CIFAR-100 across 5 runs.

| Weights | Test accuracy |
|---|---|
| Standard | $61.38\% \pm 0.0017\%$ |
| Proportional to intrinsic dimension | $\mathbf{61.77\% \pm 0.0020\%}$ |

value for independence with a $t$-test: we find that the negative correlation is significant. In other words, the higher the intrinsic dimension of a class, the harder it is to classify.

In order to check if this insight can help improve classifiers, we train the ResNet-18 in two different ways: in the first, we use the standard cross entropy loss, and in the second, we weight the terms corresponding to each class in the cross entropy loss in a manner proportional to their intrinsic dimension. The idea behind this reweighting of the loss is simply to focus more on classes of higher intrinsic dimension, as we have just shown these are harder to classify. We do not perform commonly-used data augmentation schemes so as to not change the intrinsic dimension of the data. Results are shown in Table 1, where we can see that this very simple change to the cross entropy loss marginally (but significantly, as error bars do not overlap) improves the accuracy of the network, providing an essentially "free" improvement, given the low computational overhead of estimating intrinsic dimension.

## 4. Conclusions

In this work, we have proposed and empirically verified the *union of manifolds hypothesis.* We also established that classes of higher intrinsic dimension are harder to classify, which lead us to suggest a re-weighted cross entropy loss which slightly improves classification accuracy. We hope our work will encourage the broader deep learning community to further unlock the potential of the union of manifolds hypothesis for both understanding natural data and discovering empirical improvements.

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

# Extended Abstract Track

Adam Block, Zeyu Jia, Yury Polyanskiy, and Alexander Rakhlin. Intrinsic dimension estimation. *arXiv preprint arXiv:2106.04018*, 2021.

Johann Brehmer and Kyle Cranmer. Flows for simultaneous manifold learning and density estimation. *Advances in Neural Information Processing Systems*, 33:442–453, 2020.

Michael M Bronstein, Joan Bruna, Taco Cohen, and Petar Veličković. Geometric deep learning: Grids, groups, graphs, geodesics, and gauges. *arXiv preprint arXiv:2104.13478*, 2021.

Anthony L Caterini, Gabriel Loaiza-Ganem, Geoff Pleiss, and John P Cunningham. Rectangular flows for manifold learning. In *Advances in Neural Information Processing Systems*, volume 34, 2021.

Edmond Cunningham, Adam Cobb, and Susmit Jha. Principal manifold flows. *arXiv preprint arXiv:2202.07037*, 2022.

Bin Dai and David Wipf. Diagnosing and enhancing VAE models. *ICLR*, 2019.

Ehsan Elhamifar and René Vidal. Sparse manifold clustering and embedding. In J. Shawe-Taylor, R. Zemel, P. Bartlett, F. Pereira, and K.Q. Weinberger, editors, *Advances in Neural Information Processing Systems*, volume 24. Curran Associates, Inc., 2011. URL https://proceedings.neurips.cc/paper/2011/file/fc490ca45c00b1249bbe3554a4fdf6fb-Paper.pdf.

Ehsan Elhamifar and Rene Vidal. Sparse subspace clustering: Algorithm, theory, and applications, 2012. URL https://arxiv.org/abs/1203.1005.

Ehsan Elhamifar and René Vidal. Sparse subspace clustering: Algorithm, theory, and applications. *IEEE transactions on pattern analysis and machine intelligence*, 35(11): 2765–2781, 2013.

Kaiming He, Xiangyu Zhang, Shaoqing Ren, and Jian Sun. Deep residual learning for image recognition. In *2016 IEEE Conference on Computer Vision and Pattern Recognition (CVPR)*, pages 770–778. IEEE, 2016.

Konik Kothari, AmirEhsan Khorashadizadeh, Maarten de Hoop, and Ivan Dokmanić. Trumpets: Injective flows for inference and inverse problems. In *Proceedings of the Thirty-Seventh Conference on Uncertainty in Artificial Intelligence*, volume 161, pages 1269–1278, 2021.

Alex Krizhevsky, Geoffrey Hinton, et al. Learning multiple layers of features from tiny images. 2009.

Yann LeCun, Léon Bottou, Yoshua Bengio, and Patrick Haffner. Gradient-based learning applied to document recognition. *Proceedings of the IEEE*, 86(11):2278–2324, 1998.

Elizaveta Levina and Peter Bickel. Maximum likelihood estimation of intrinsic dimension. *Advances in neural information processing systems*, 17, 2004.

Uzu Lim, Vidit Nanda, and Harald Oberhauser. Tangent space and dimension estimation with the wasserstein distance. *arXiv preprint arXiv:2110.06357*, 2021.

Gabriel Loaiza-Ganem, Brendan Leigh Ross, Jesse C Cresswell, and Anthony L. Caterini. Diagnosing and fixing manifold overfitting in deep generative models. *Transactions on Machine Learning Research*, 2022. URL https://openreview.net/forum?id=0nEZCVshxS.

David JC MacKay and Zoubin Ghahramani. Comments on'maximum likelihood estimation of intrinsic dimension'by e. levina and p. bickel (2004). *The Inference Group Website, Cavendish Laboratory, Cambridge University*, 2005.

Yuval Netzer, Tao Wang, Adam Coates, Alessandro Bissacco, Bo Wu, and Andrew Y Ng. Reading digits in natural images with unsupervised feature learning. 2011.

Phil Pope, Chen Zhu, Ahmed Abdelkader, Micah Goldblum, and Tom Goldstein. The intrinsic dimension of images and its impact on learning. In *International Conference on Learning Representations*, 2021.

Brendan Leigh Ross and Jesse C Cresswell. Tractable density estimation on learned manifolds with conformal embedding flows. In *Advances in Neural Information Processing Systems*, volume 34, 2021.

Brendan Leigh Ross, Gabriel Loaiza-Ganem, Anthony L Caterini, and Jesse C Cresswell. Neural implicit manifold learning for topology-aware generative modelling. *arXiv preprint arXiv:2206.11267*, 2022.

Olga Russakovsky, Jia Deng, Hao Su, Jonathan Krause, Sanjeev Satheesh, Sean Ma, Zhiheng Huang, Andrej Karpathy, Aditya Khosla, Michael Bernstein, et al. Imagenet large scale visual recognition challenge. *International journal of computer vision*, 115(3):211–252, 2015.

Karen Simonyan and Andrew Zisserman. Very deep convolutional networks for large-scale image recognition. *ICLR*, 2015.

Piotr Tempczyk, Rafał Michaluk, Lukasz Garncarek, Przemysław Spurek, Jacek Tabor, and Adam Golinski. Lidl: Local intrinsic dimension estimation using approximate likelihood. In *International Conference on Machine Learning*, pages 21205–21231. PMLR, 2022.

Han Xiao, Kashif Rasul, and Roland Vollgraf. Fashion-MNIST: a novel image dataset for benchmarking machine learning algorithms. *arXiv preprint arXiv:1708.07747*, 2017.

## Appendix A. Verifying the union of manifolds hypothesis

We show a more granular breakdown of Figure 1, with intrinsic dimension estimate values for each class in Figure 3 for MNIST, Figure 4 for FMNIST, Figure 5 for SVHN, and Figure 6 for CIFAR-10. Since CIFAR-100 and ImageNet have too many classes to show, we only include the top 5 highest and lowest intrinsic dimension classes in Figure 7 for CIFAR-100, and Figure 8 for ImageNet.

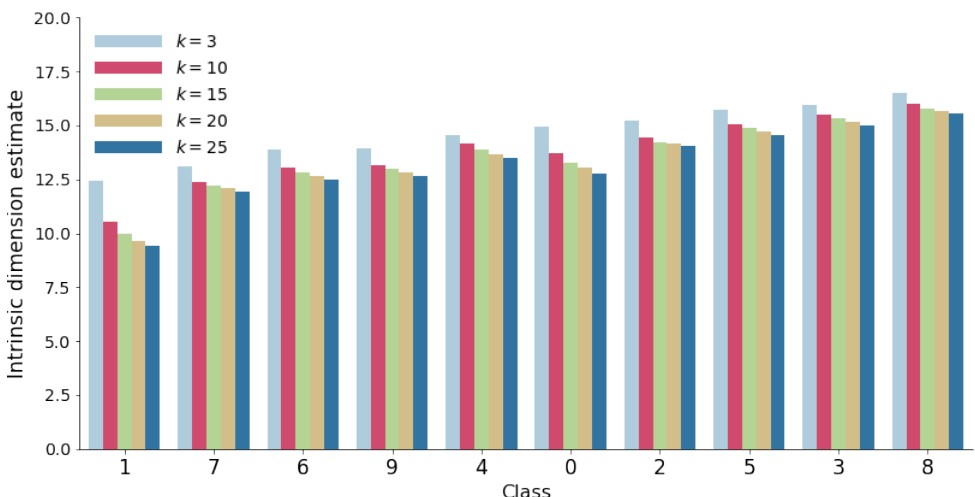

Figure 3: Intrinsic dimension estimates for classes in the MNIST dataset.

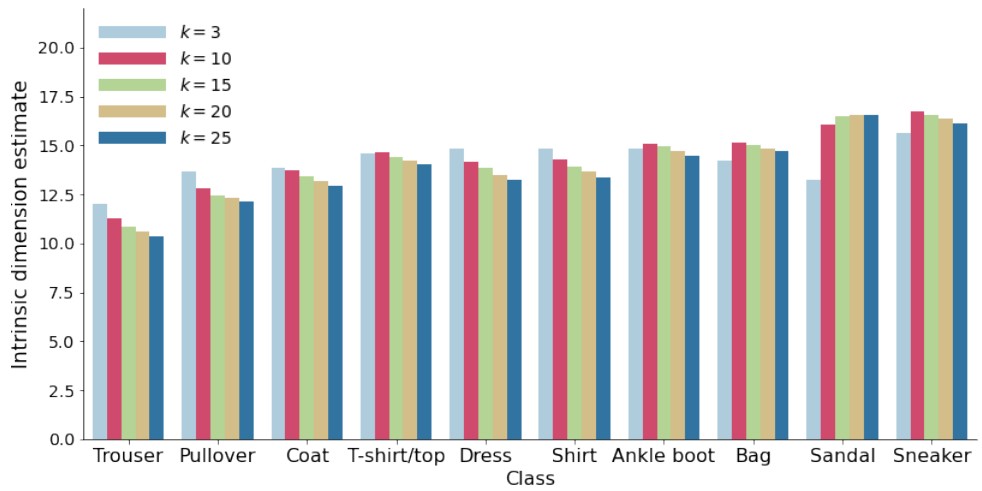

Figure 4: Intrinsic dimension estimates for classes in the FMINST dataset.

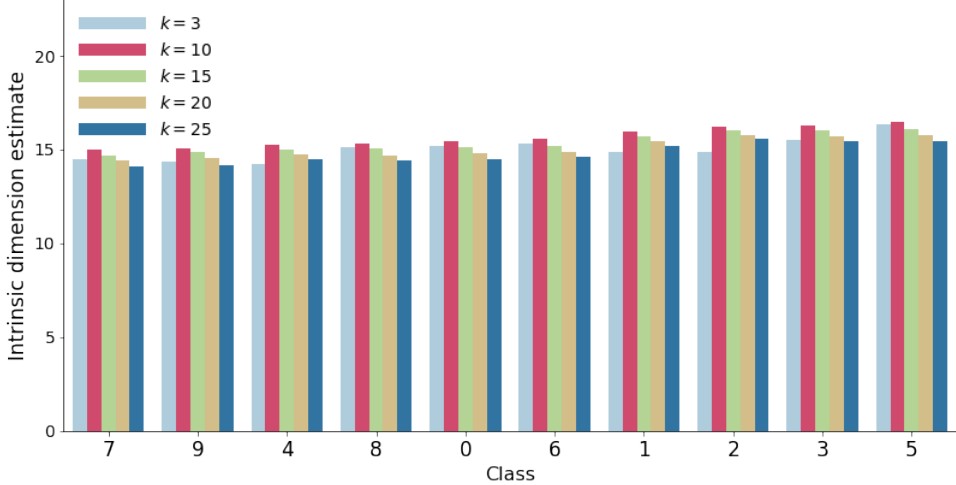

Figure 5: Intrinsic dimension estimates for classes in the SVHN dataset.

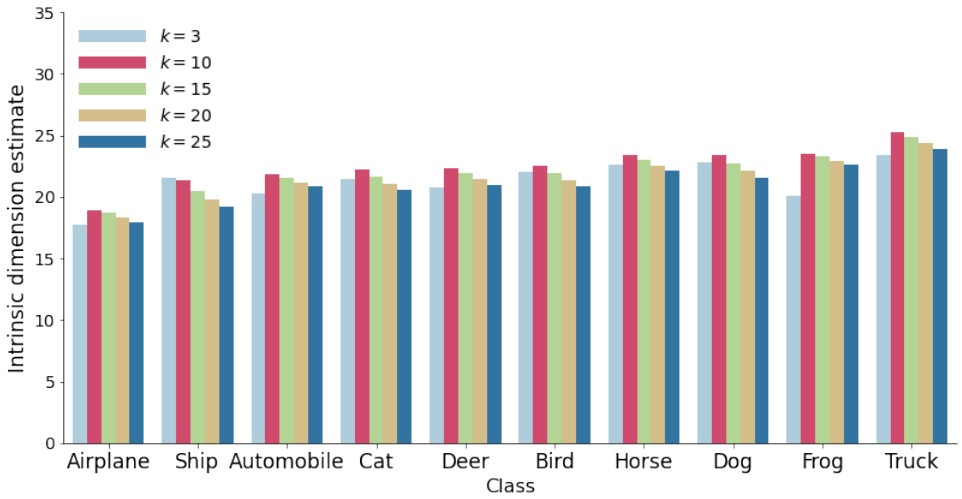

Figure 6: Intrinsic dimension estimates for classes in the CIFAR-10 dataset.

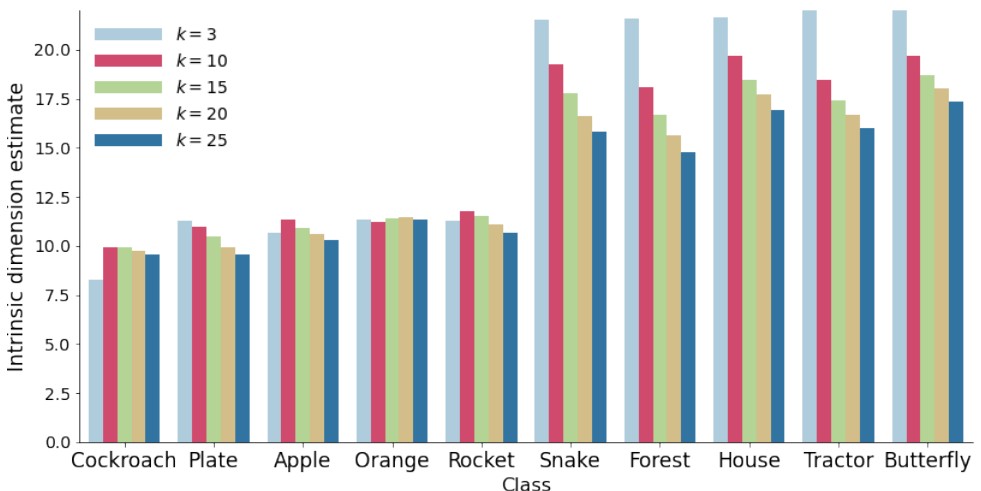

Figure 7: Intrinsic dimension estimates for classes in the CIFAR-100 dataset.

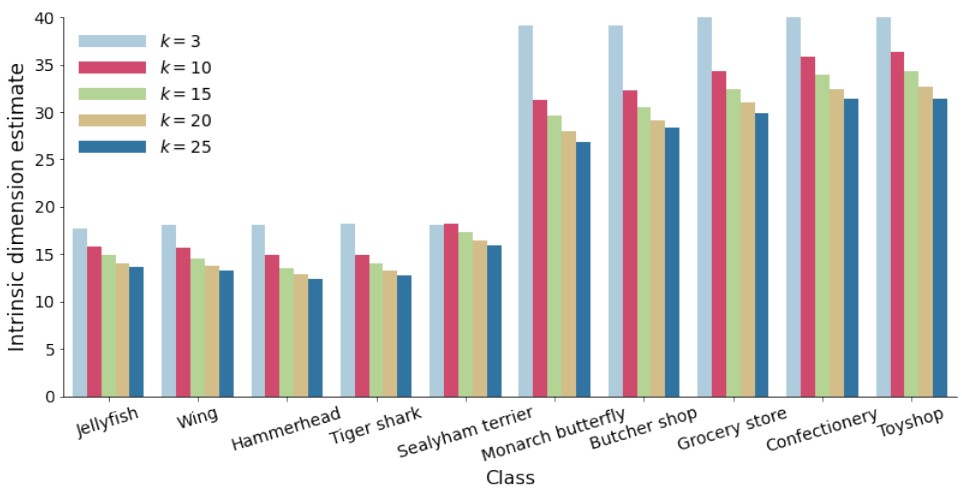

Figure 8: Intrinsic dimension estimates for classes in the ImageNet dataset.

Extended Abstract Track

Extended Abstract Track