# OpenReview forum: "The Union of Manifolds Hypothesis"
_NeurIPS.cc/2022/Workshop/NeurReps — NeurReps 2022 Poster_

### Official Review · Reviewer_3FJw · 2022-10-10
**Review for `The Union of Manifolds Hypothesis'**

**Confidence:** 4
**Soundness:** 2
**Presentation:** 4
**Contribution:** 2
**Overall Rating:** 4

**Summary:**

The authors propose that the community starts viewing data as lying on a union of manifolds, rather than a manifold, with the difference being that the different connected components are modeled as having distinctly different intrinsic dimensions.

To that end, the authors propose classes in a dataset as a natural divisor for the connected components. To test their hypothesis, they compute the intrinsic dimensions for classes in many well known datasets and showcase their range.

They then argue about the benefits on the hypothesis on the classification accuracy of predictors, when classes are normalized using their intrinsic dimension, and show relevant experiments.

**Questions:**

The most pressing question is why exactly do we need to consider a union of manifolds model? Either there needs to be a very convincing theoretical answer for this, or extensive experimentation to showcase that it is purely better than simply using the overall intrinsic dimension. I am also confused as to the limitation, on a theoretical level. The mobius strip has an intrinsic dimension of 2, however it also contains its boundary which has a lower intrinsic dimension. On a theoretical level, why is it a limitation to consider the higher dimension as the intrinsic dimension, if known manifolds can circumvent this by having different intrinsic dimensions in different parts?

Less of a question but more of a suggestion: in Figure 1, the axis label implies that this is the range (max value - min value) of the intrinsic dimension, whereas this is not what's plotted. In fact, it's not clear what is actually plotted: is it the mean dimension? The figure should be more clear about what is being shown.

Finally, how can the intrinsic dimension of the whole dataset be lower than the maximum intrinsic dimension of the classes? For that to be true it would imply that there is overlap between the manifolds of the different classes, bringing the hypothesis into question.

**Limitations:**

Other than what has already been discussed, the main limitation of the paper is the experimental evaluation. Again, while this shouldn't be a dealbreaker for a workshop paper, when the paper has no theoretical contributions and its content revolves a hypothesis, that hypothesis has to be evaluated.

To be concrete, Figure 2 is not convincing. There seem to be hardly any trends and the data is very noisy: I'm not sure convincing takeaways can be deduced from these experiments. Moreover, the accuracy gain is only tested on CIFAR-100. While I don't necessarily doubt the reasoning for that (i.e. that other datasets are simple and their estimates were less reliable), it is trying to deduce a trend from a single data point (which itself is not very convincing as it shows a truly marginal improvement). While other datasets might be simple, the authors could test the performance of simple, dense classifiers who do not achieve state of the art performance on these tasks and evaluate if their hypothesis leads to improved results.

**Recommended Decision:**

2: Borderline

**Relevance:**

3: Solid fit

**Strengths And Weaknesses:**

Originality: The hypothesis of the authors is moderately original, to the best of my knowledge. However, it is a very obvious observation: a dataset is an arbitrary collection of training data belonging to different semantic classes and there is no way the intrinsic dimension of all those classes would be the same.

Quality: The submission is technically sound, however the experiments are not extensive. While this is expected for a workshop submission, as I stated, the novelty of the submission is fairly limited, which translates to the fact that experiments need to be very convincing in order for people to accept that there is benefit in the approach. We could consider that the intrinsic dimension of the manifold is the maximum of all the intrinsic dimensions: the experiments need to clearly show that doing so, versus computing the intrinsic dimension of every class, is significantly inferior.

Clarity: The paper is very clear, well written, and pleasant to read.

Significance: The paper is of interest to this community. Many people are thinking of datasets in terms of manifolds these days, and any advance in our understanding is very welcomed.

**Submission Track:**

Extended Abstract (4 Page)

---

### Official Review · Reviewer_pvHX · 2022-10-14
**The Union of Manifolds Hypothesis**

**Confidence:** 4
**Soundness:** 4
**Presentation:** 4
**Contribution:** 4
**Overall Rating:** 10

**Summary:**

The authors propose an alternative to the classical manifold hypothesis: the union of manifolds hypothesis, stating that the data lie in several manifolds of different intrinsic dimensions. They identify these manifolds as the data classes, they estimate the dimensions of these manifolds and the results showing different intrinsic dimensions support the union of manifolds hypothesis. To improve classifiers, the authors propose to increase the high-dimensional classes in the cross entropy loss to give more credit to the low-dimensional classes for which the classification is more accurate.

**Questions:**

The results strongly depend on formula (1). Could you briefly explain the intuition behind this formula? Are there other estimators of the dimension that could be used to strengthen the results?

You spend some time to argue that "the higher the intrinsic dimension of a class, the harder it is to classify". This statement seems quite natural. Isn't it simply due to the curse of dimensionality?

There is a type of "unions of manifolds" that is getting interest in the community of geometric statistics, which are stratified spaces. The connected components are glued together in a certain way like the cube (open cube + faces + edges + vertices) or the cone of symmetric positive semi-definite matrices for instance. This may be the adapted overall structure on the union of manifolds. As you say in your comment on 9s and 3s, the connections between manifolds may be spaces that do not represent elements of the class. However, having one (low-dimensional) stratified space that contains all the digits might be practical, for example to efficiently compute distances. Could you maybe connect your topic to the one of stratified spaces?

**Limitations:**

The authors prevent some limitations by testing several hyperparameters, several datasets. The discussions are sound and the authors don't pretend more than exploring the validity of a new hypothesis.

**Recommended Decision:**

3: Accept

**Relevance:**

4: Highly relevant

**Strengths And Weaknesses:**

The novelty is the first empirical verification of the union of manifolds hypothesis, which is very appealing. The submission is technically sound and clear. The results are promising and are of interest for the community.

**Submission Track:**

Extended Abstract (4 Page)

---

### Official Review · Reviewer_9bcr · 2022-10-19
**Assuming multiple dimensionality classes improves classification performance**

**Confidence:** 4
**Soundness:** 3
**Presentation:** 4
**Contribution:** 3
**Overall Rating:** 8

**Summary:**

The authors propose a 'Union of Manifolds' hypothesis - that high-dimensional data often arise from the union of discrete manifolds. They show this by characterizing the intrinsic dimensionality of image datasets, and show that there is essentially a free improvement in classification to be gained by weighting by dimensionality.

They identify connected components in various image datasets and use a previously established technique to quantify the intrinsic dimensionality (ID) of each connected component. The variance in ID is taken as support of the existence of multiple manifolds. Results are consistent across choices of the hyperparameter in the ID computation.

Considering class labels are a proxy for images of similar dimensionality, the authors show that there is an inverse relationship between classification accuracy and ID.

Weighting cross-entropy loss by the ID results in a marginal but significant improvement in RESNET-18 classification accuracy.



**Questions:**

Fig. 1: The authors mention that k is a hyperparameter - but what does it signify?

Page 2 footnote: "The disjoint union of d-dimensional manifolds is itself a d-dimensional manifold: the possibility of having different intrinsic dimensions differentiates the union of manifolds hypothesis from the manifold hypothesis." This is an important point that should be in the main text.

**Limitations:**

No major limitations, suggestions are included in the previous section.

**Recommended Decision:**

3: Accept

**Relevance:**

3: Solid fit

**Strengths And Weaknesses:**

Originality: There has been other work that shows that datasets can be encoded by a union of manifolds. However, this abstract is original in that it shows a specific classification performance improvement from assuming this hypothesis a priori.

Quality: The submission is technically sound.

Claims: The claims are reasonably well supported, apart from the weakness I describe below.

Clarity: Very well organized and clearly written. I appreciate the authors providing sentences that provide an intuition for the concepts discussed, e.g. talking about similar images morphing into one another.

Weaknesses: My biggest gripe is about objectivity in your assertions about Fig. 1. While I agree that variability in intrinsic dimensionality (ID) is in support of the hypothesis, what constitutes a significant variance? In a later submission, I would hope that you address this through statistical tests, and examining the variance within each class. Is the variance of each class related to the ID itself - in other words, is there a signal-dependent noise?

**Submission Track:**

Extended Abstract (4 Page)

---

### Decision · Program_Chairs · 2022-10-21

Accept (Poster)